# Heterogeneity in Melanoma

**DOI:** 10.3390/cancers14123030

**Published:** 2022-06-20

**Authors:** Mei Fong Ng, Jacinta L. Simmons, Glen M. Boyle

**Affiliations:** 1Cancer Drug Mechanisms Group, QIMR Berghofer Medical Research Institute, Brisbane, QLD 4006, Australia; meifong.ng@qimrberghofer.edu.au (M.F.N.); j6.simmons@qut.edu.au (J.L.S.); 2School of Biomedical Sciences, Faculty of Health, Queensland University of Technology, Brisbane, QLD 4000, Australia; 3School of Biomedical Sciences, Faculty of Medicine, University of Queensland, Brisbane, QLD 4072, Australia

**Keywords:** melanoma, tumour heterogeneity, intertumoural heterogeneity, intratumoural heterogeneity, resistance

## Abstract

**Simple Summary:**

Tumour heterogeneity is a phenomenon where the cancer cells evolve diversely over the course of the disease. As a result of the evolution, the cancer cells can be found to be genetically, epigenetically and/or phenotypically different in order to survive in the human body. The tumour microenvironment also plays a crucial role during the evolution. This review compiles the recent developments in melanoma tumour heterogeneity and how heterogeneity impacts the evolution of the disease. The recent findings have greatly improved our knowledge on heterogeneity in melanoma, but there remain many unanswered questions.

**Abstract:**

There is growing evidence that tumour heterogeneity has an imperative role in cancer development, evolution and resistance to therapy. Continuing advancements in biomedical research enable tumour heterogeneity to be observed and studied more critically. As one of the most heterogeneous human cancers, melanoma displays a high level of biological complexity during disease progression. However, much is still unknown regarding melanoma tumour heterogeneity, as well as the role it plays in disease progression and treatment response. This review aims to provide a concise summary of the importance of tumour heterogeneity in melanoma.

## 1. Introduction

There were an estimated 325,000 new cases of melanoma diagnosed worldwide in 2020, causing an economic burden and placing enormous pressure on healthcare systems [1]. Cutaneous melanoma arises when melanocytes, the cells that produce melanin and provide ultraviolet (UV) protection to the skin [2,3], undergo malignant transformation. The transformation of melanocytes to melanoma has been extensively described [4]. Melanoma may arise from single melanocytes or clusters of melanocytes known as naevi, with or without *BRAF* or *NRAS* mutation [5,6,7,8]. Vast evidence has suggested that exposure to UV radiation results in mutations in DNA, in particular UV-induced C > T transitions that are implicated in melanoma development [9,10]. Both UVA and UVB radiation can induce DNA damage responses through production of pyrimidine dimers and generation of oxidative responses, although at different rates via different mechanisms [11]. Various genetic factors are correlated with the progression of melanoma. In general, melanoma can be classified into four subtypes based on the mutation carried in the tumour, i.e., *BRAF*, *NRAS*, *NF1* or triple wild type [4]. Approximately 50% of all cutaneous melanoma cases exhibit *BRAF^V600E^* mutations, while *NRAS* mutations are found in up to 20% of the cases [12,13]. *TERT* promoter mutations have been detected in intermediate lesions, while *NF1* mutation is observed in melanoma in situ [14,15]. In the advanced stages of melanoma, mutations in *CDKN2A*, *PTEN* and *TP53* were observed [15]. Melanoma has the highest mutation frequency among human cancers [16,17,18,19]. The increasing mutational burden as the disease progresses is thought to contribute to increased heterogeneity. Heterogeneity in all cancers, including haematological and solid tumours, is now a well-recognised barrier, particularly in looking for the best treatment for cancer patients [20]. In general, heterogeneity can be categorised into interpatient heterogeneity, intertumoural heterogeneity and intratumoural heterogeneity (Figure 1A,B). While interpatient heterogeneity describes the differences in tumours between patients, intertumoural heterogeneity refers to differences between lesions in the same patient. It is now known that populations of cancer cells within a given tumour may also be different from each other, described as intratumoural heterogeneity. Melanoma tumour heterogeneity has been reviewed previously [21,22]. In this review, we provide an update on the development in melanoma intertumoral and intratumoural heterogeneity.

## 2. Intertumoural Heterogeneity

Advancements in molecular biology and sequencing techniques have made detection of intertumoural heterogeneity easier. Sakaizawa and colleagues investigated mutational diversification between circulating tumour cells in patients where paired primary and metastatic lesions were available. Their data revealed discrepancies in the *BRAF* mutation status in 33% (3/9) of the paired samples and *KIT* mutations in 75% (3/4) [23]. Similarly, another study showed that mismatched *BRAF/NRAS* mutations were found in 15% (15/99) of the paired primary and metastatic lesions [24]. This discordant *BRAF* mutation status has similarly been identified in multiple studies and poses a significant problem for treatment selection and response (see Section 5) [25,26,27,28,29,30]. Moreover, differential methylation was detected between three paired primary and metastatic cell lines, where at least 7220 differential methylation fragments were identified [31]. Subsequent data revealed that *EBF3* promoter methylation causes elevated *EBF3* expression, which leads to metastatic characteristics in melanoma cells. Aberrant expression of methyltransferases in melanoma has also been proposed as a mechanism of malignant progression [32]. More detailed studies have subsequently been conducted to understand the phylogenetic relationship between the primary tumour and metastases. The work demonstrated that metastases are genetically diverse in comparison to the primary tumour or other metastases from the same patient. Whole-genome sequencing (WGS) analysis indicated that metastases can emerge from either identical or distinct cell populations from the primary tumour [33,34]. In addition to mutations derived from primary tumours, Ding and colleagues also identified additional mutations that were detectable in metastases but not primary tumours [19]. Notably, intertumoural heterogeneity was observed in 33 tumour samples from 15 patients, where only ~50% non-synonymous exonic mutations were shared between synchronous metastases [35]. In a study of paired primary and metastatic samples, Mejbel and colleagues found at least one mutation in each of the three patients that was heterogeneously expressed between the primary and metastatic sample [36]. They also showed that paired primary and metastatic lesions may be morphologically distinct, consistent with findings at the molecular level. This is intriguing as histomorphology could be used to categorise heterogeneous tumours; however, more clarity is required as the sample size (*n* = 3) in this study was relatively small. Further, in a case where a total of 13 metastases were extracted from a treatment-naïve patient, multiple mutation clusters and clear lineage separation were observed [34]. Additionally, aneuploidy and whole-genome doubling appear to occur at a higher frequency in late-stage samples compared to matched primary tumour samples [37].

Overall, it is conceivable that the metastases are subpopulations disseminated from a primary tumour or tumour cells that have gone through further adaptations and/or selection to become genetically or phenotypically distinct from the primary tumour or a regressed primary tumour. It is clear that mutations accumulate throughout the development of metastases and metastases within a single patient do not share the same mutations. Cancer cells gain mutations, resulting in additional mutational clusters, hyperploidy and whole-genome doubling, whereas cancer cells also lose mutations, causing a loss in heterozygosity. This intertumoural divergence, or better described as intertumoural evolution, likely enhances survival and promotes metastasis.

## 3. Intratumoural Heterogeneity

Intratumoural heterogeneity describes the differences between cells or subpopulations of cells within a tumour. Heterogeneous cells within a tumour can display remarkable variability in phenotypic traits (Table 1). A cross-sectional analysis that screened 1165 tumour exomes from 12 different cancers revealed that intratumoural heterogeneity was found in all cancer types, with 86% of the samples consisting of at least two subpopulations, with melanoma being the most heterogeneous [18]. Furthermore, a study has shown that 6/9 (67%) melanoma tumours exhibited intratumoural heterogeneity [25].

### 3.1. Genetic Contribution to Intratumoural Heterogeneity

Genetic instability may cause mutations to be more prevalent, leading to the existence of subpopulations with variable phenotypic traits within a tumour [38]. A study has identified at least two subpopulations in 11/15 (73%) metastatic melanoma tumours via WGS. Using loss of heterozygosity and copy number variations as references and associating mutations with specific populations, the group discovered that the founding population and secondary populations in the same tumours displayed different genetic variations [19]. Interestingly, the founding population in one particular tumour exhibited a UV damage mutational profile, while this was not the case for subpopulation [19]. The authors speculated that defective DNA repair may have taken place in the subpopulation, resulting in a significantly increased mutational burden. This study demonstrated that intratumoural heterogeneity is actively acquired in favour of melanoma progression.
cancers-14-03030-t001_Table 1Table 1Summary of three different aspects contributing to intratumoural heterogeneity.HeterogeneitySubpopulationsReferencesGenetic*BRAF* wild type and *BRAF* mutants[25,39,40]
Heterogenous expression of BRAF^V600E^[41]
*KIT^WT^* and *KIT^L576P^*[42]
*BRAF^V600E^/NRAS^WT^* and *BRAF^WT^/NRAS^Q61R^*[43]
*NRAS^wild type^* and *NRAS^G13R^*[44]Epigenetic*RASSF1A*, *CDKN2A*, *DAPK*, *MGMT* and *RB1*[45]
Differential methylation leads to heterogeneous expressions of MAGE-A3[46]
H3K27 hypermethylation[47]
JARID1B+ and JARID1B−[48,49]PhenotypicMITF^high^ and MITF^low^[50,51,52,53,54,55,56,57,58,59]
MITF and BRN2[60,61,62,63,64,65,66,67]
MITF and PAX3[66]
Transition from MITF^high^/NF-κB^low^ to MITF^low^/NF-κB^high^/AXL^high^ during acquisition of resistance[68,69]
Transition from ZEB2^high^/SNAIL2^high^/ZEB1^low^/TWIST1^low^ to ZEB2^low^/SNAIL2^low^/ZEB1^high^/TWIST1^high^ in primary melanoma to metastatic melanoma[70]
ABCB5+ and ABCB5-[71,72,73,74]
CD133+ and CD133-[74,75,76,77,78]
NGFR+ and NGFR-[79,80]
Transition from MART-1^neg^/NGFR^high^ to MART-1^neg^/NGFR^neg^ upon BRAFi treatment[81]
ALDH+ and ALDH-[82]
NME1^high^ and NME1^low^[83,84]
PGC1α^high^ and PGC1α^low^[85,86]
MCT1^high^ and MCT1^low^[87]


Genetic heterogeneity in melanoma can be explored by examining allelic variations using DNA microsatellite loci analysis [88,89]. Alternatively, analyses, such as sequencing, high-resolution melting analysis, immunohistochemistry (IHC) and mutation-specific PCR, have also been utilised to reveal the presence of cells with either wild type or mutant copies of BRAF in the same lesion/metastasis [25,39,40]. Busam and colleagues found heterogeneous BRAF^V600E^ staining as determined by IHC analysis with a mutation-specific antibody in 27% (6/22) of the genetically confirmed BRAF^V600E^-positive cases [41], suggesting either heterogeneity of the BRAF mutation in the lesions or a lack of protein expression. In a case study, subpopulations carrying *KIT^WT^* and its mutant *KIT^L576P^* were identified in circulating DNA of a metastatic vaginal mucosal melanoma patient [42]. While it is common to find multiple mutations within a tumour, mutations that are mutually exclusive in different tumour cell populations within a single tumour have also been identified. Sensi and colleagues discovered that 75% (75/100) of the clones of a short-term melanoma cell line expressed *BRAF^V600E^/NRAS^WT^*, while the remaining expressed *BRAF^WT^/NRAS^Q61R^* [43]. Subsequent investigations revealed that the two subpopulations of cells exhibit different phenotypes. For example, while the *BRAF^V600E^/NRAS^WT^* cells remained viable under an ultra-low attachment condition, the *BRAF^WT^/NRAS^Q61R^* cells were more proliferative and tumourigenic. Further, subpopulations *NRAS^WT^* and *NRAS^G13R^* were found in the same BRAFi-resistant tumour in a melanoma patient [44]. In a sophisticated study, a metastatic lesion from a patient that had undergone multiple treatment regimens was divided into several regions in which deep targeted DNA sequencing was performed [90]. The data showed that only 28% (15/53) of the somatic mutations identified were prevalent in all 41 regions sequenced, while 30% of the mutations were confined to a single region. Further, copy number alteration analysis not only revealed modifications, such as gains of chromosome 6p and 20q and losses of chromosome 6q and 9p, that happened across all 41 regions but also alterations that only occurred in certain regions, such as chromosome 7 gain in four regions, whole-chromosome 10 loss in five regions, 10p loss in one region and chromosome 13 gain in four regions. Taken together, there are large amounts of genetic heterogeneity in melanoma, resulting in subpopulations within a tumour that can possess certain survival advantages and, therefore, possibly affect the treatment outcome.

### 3.2. Intratumoural Heterogeneity from Non-Genetic Sources

#### 3.2.1. Epigenetic Heterogeneity

Epigenetic mechanisms can contribute to melanoma development via DNA methylation, chromatin remodelling, such as histone modification, and non-coding RNA regulation [91]. Changes in methylation status between subpopulations of cells in melanoma tumours have been observed across multiple studies. For instance, heterogeneous hypermethylation of at least one of the promoters of various tumour-related genes (*RASSF1A*, *CDKN2A*, *DAPK*, *MGMT* and *RB1*) was observed in 70% (16/23) of the patient samples [45]. Of note, expression of RASSF1A was observed at the periphery but not in the core of the tumours, indicating that varied degrees of hypermethylation can occur in different areas of a tumour. Differential methylation was observed in clones derived from a melanoma lesion, which then resulted in heterogeneous expressions of MAGE-A3, a previously identified immunotherapeutic target in melanoma [46]. This could dampen the therapeutic efficacy in melanoma patients. In addition, H3K27 hypermethylation in the core of solid tumours caused by low glutamine can lead to cell dedifferentiation and reduced sensitivity to BRAF inhibitor treatment [47]. JARID1B, a member of the jumonji/ARID1 (JARID1) histone 3 K4 demethylase family, was expressed in a slow-cycling population and identified as being essential for tumour maintenance and metastatic progression [48]. Roesch and colleagues also found that melanoma cells maintain heterogeneous expressions of JARID1B. Pure JARID1B+ or JARID1B− cell populations were generated by cell sorting; however, the populations became heterogeneous for JARID1B after just 72 h in culture, showing that expression of JARID1B is dynamically regulated [49]. Epigenetic alterations are also often associated with drug resistance. Shaffer and colleagues found more than 10,000 differentially accessible sites from the assay for transposase-accessible chromatin with sequencing (ATAC-seq) by comparing pre-resistant cells to drug-induced resistant cells [92], indicating that drug treatment promotes epigenetic alterations, which lead to treatment resistance and cancer cell survival.

#### 3.2.2. Phenotypic Heterogeneity

It is possible for a tumour to display dynamic expression of certain markers in different regions in response to microenvironmental cues, disease development, therapeutic treatment or progression without going through genetic alterations. This has been described as phenotypic heterogeneity. For instance, a single human melanoma cell can grow into a heterogeneous tumour in an immunocompromised mouse [93]. In addition, heterogeneous expression of various markers (such as ABCB5, NGFR, CD54, KIT, etc.) has been identified in tumour samples, although there are no significant differences in the tumourigenicity or metastatic properties of the subpopulations with differing expression [93,94]. Hoek and colleagues initially proposed a reversible proliferative and invasive model based on their observations [95], which coincides with the occurrence of heterogeneity in melanoma. Many studies have identified gene signatures associated with reversible proliferative or invasive states in melanoma, where genes that were normally expressed in the proliferative state were often inversely correlated in cells in the invasive state and vice versa [50,96,97]. Remarkably, cells in a proliferative state have been shown to behave similarly to invasive cells when the two populations were co-cultured together [51,98]. Subsequent findings revealed that heterogeneous populations of proliferative and invasive cells are more metastatic and invasive compared to single proliferative populations [51,98], suggesting cooperation between heterogeneous populations in driving metastasis.

Heterogeneous expression of microphthalmia-associated transcription factor (MITF) has been observed extensively in melanoma development and progression. Importantly, MITF^high^ and MITF^low^ subpopulations co-exist in different regions of patient tumour samples [52,53,54,55]. While MITF^high^ expression is known to promote differentiation and proliferation, MITF^low^ expression is associated with an invasive and stem-cell-like phenotype [50,56]. The interaction between MITF^high^ and MITF^low^ subpopulations was elegantly shown in a zebrafish melanoma xenograft model. When invasive MITF^low^ (WM266-4) cells and less invasive MITF^high^ (501mel) cells were co-injected into zebrafish embryos, the MITF^high^ cells phenocopied the MITF^low^ cells without changing the MITF expression and became invasive [51]. When the mixed populations of MITF^low^ (75%) and MITF^high^ (25%) were co-injected into mice, the proliferation rate of the tumour was higher compared to single populations and the MITF^high^ population increased from 25% to 50% [57]. Subsequent RNA sequencing results revealed that even the heterogeneous tumours expressed higher MITF as compared to the MITF^low^ tumours; the tumours were enriched for markers of epithelial mesenchymal transition (EMT) and more invasive, less proliferative phenotypes, showing that there is phenotype adaptation that happens due to cooperativity between MITF^high^ and MITF^low^ cells in enhancing metastasis. Notably, a MITF^low^ zebrafish model has been established to address a subgroup of melanoma patients with low MITF having a poor prognosis. The researchers first incorporated a *BRAF^V600^* mutation into the MITF^low^ zebrafish and found that the zebrafish developed nevi and melanoma [58]. Subsequently, a *TP53* mutation was introduced into the model and the zebrafish first developed black pigmented lesions that then progressed into superficial and nodular melanoma [59]. As the disease progressed, the expression of MITF shifted from low levels in superficial melanoma to high levels in nodular melanoma. Interestingly, by integrating *BRAF^V600^* and/or *TP53* mutation(s) in the MITF^low^ zebrafish model, melanoma onset is triggered and, over time, the disease progresses by alternating MITF expression. Collectively, this evidence has shown that MITF is heterogeneously expressed in tumours, and the role of MITF in promoting melanoma heterogeneity cannot be disregarded.

The co-existence of the MITF^high^ subpopulation with other subpopulations can result in melanoma progression. BRN2, a transcription factor encoded by the gene *POU3F2*, was found heterogeneously expressed in mouse xenografts [60]. BRN2^high^ cells were migratory and less pigmented, suggesting they potentially expressed low levels of MITF. Intriguingly, MITF and BRN2 were found mutually exclusively expressed in two distinct subpopulations of cells [61,62,63]. Mechanistically, BRN2 repression is mediated by miR-211, which itself is under transcriptional control of MITF [64]. Conversely, BRN2 can promote melanoma invasion via repressing MITF expression by binding to the MITF promoter or through activation of Notch signalling, followed by activation of miR-222/221 promoter [61,62,65]. Heterogeneous expression of PAX3 in melanoma also contributes to MITF regulation by increasing binding of BRN2 to the MITF promoter [66]. When cells are BRN2^high^/PAX3^low^, the MITF expression is mainly regulated by BRN2, while, when BRN2^low^/PAX3^high^ in cells, the MITF expression is driven by PAX3. Additionally, MITF and BRN2-expressing cells act jointly in melanoma progression by first stimulating tumour growth with MITF^high^ cells and then activating tumour invasion with BRN2^high^ cells [63]. Moreover, expression of both MITF and BRN2 proved indispensable for melanoma to efficiently metastasise [67]. Overall, the complex interactions between MITF and BRN2 are essential in regulating each other, creating a heterogeneous tumour and in driving melanoma progression.

The expression level of MITF may be associated with melanoma cell behaviour in conjunction with other factors. Of note, MITF^high^/NF-κB^low^ cells can transition to MITF^low^/NF-κB^high^/AXL^high^ cells during acquisition of resistance, suggesting the importance of low MITF expression in treatment resistance [68,69]. Heterogeneous expression of AXL is commonly observed in melanomas [99]. It has been suggested that AXL has a pertinent role in the development of acquired resistance to BRAFi and MEKi. Similarly, a shift from ZEB2^high^/SNAI2^high^/ZEB1^low^/TWIST1^low^ primary melanoma to ZEB2^low^/SNAI2^low^/ZEB1^high^/TWIST1^high^ metastatic disease was observed in patient samples [70]. Follow-up experiments suggested that ZEB2 and SNAI2 act as tumour suppressors but ZEB1 and TWIST1 facilitate BRAF signalling in melanoma transformation by downregulating MITF. Thus, a loss of MITF not only correlated with treatment resistance but also melanoma metastasis.

A subpopulation of melanoma cells has been described as slow-cycling due to a slower proliferation rate than other cells under the same conditions. The population, however, could be much more aggressive or invasive compared to other cells due to expression of ABCB5, a transmembrane glycoprotein from the ATP-binding cassette (ABC) superfamily [71,72]. In the study, the ABCB5+ subpopulation exhibited self-renewing and differentiating phenotypes in addition to playing significant roles in melanoma growth, tumourigenicity and metastasis [71,72]. The data also showed that the ABCB5+ subpopulation included slow-cycling cells and was potentially involved in drug resistance mechanisms [73]. Furthermore, a stem cell marker that is frequently found co-expressed with ABCB5, CD133, was also expressed in tumourigenic, tumour-initiating cells [75,76]. CD133+ cells were also more proliferative, able to survive ultra-low attachment conditions and have an immunosuppressive phenotype compared to the CD133− subpopulation in a mouse xenograft model [76,77]. Moreover, CD133+ cells are invasive and metastatic when tested in both in vitro and in vivo (zebrafish and athymic mice) systems [78]. Evidence has shown that ABCB5- and CD133-expressing cells are co-localised in the perivascular niche where vasculogenic mimicry takes place in melanoma xenografts. This suggests the involvement of ABCB5+ and CD133+ cells in tumour growth and progression by promoting vasculogenic mimicry and morphogenesis of tumour microcirculation [74]. A NGFR+ subpopulation has also been identified as cancer initiating cells due to their capability of forming tumours and metastasising in mice compared to NGFR− subpopulations [79,80]. Interestingly, only NGFR+ xenografts formed in nude and NOD/SCID mice, not NSG mice, were able to phenocopy parental tumours. This was contrary to earlier work demonstrating no change in the tumourigenic potential of cells on the basis of ABCB5, CD133 or NGFR expression [93,94]. However, changes in the expression of these proteins during tumour formation in mice is unknown, and, therefore, it is possible that heterogeneous populations are regenerated as tumours develop. Nonetheless, Su and colleagues observed a trajectory of phenotypic changes in patient-derived cell lines upon BRAFi treatment from day 0 until a stably drug-resistant population is established [81]. The cells initially expressed MART-1, a melanocytic marker, then transitioned into a slow-cycling (MART-1^neg^/NGFR^high^) phenotype after 2 weeks, followed by a mesenchymal (MART-1^neg^/NGFR^neg^) phenotype at day 62. This has showcased the phenotypic changes in a process of developing resistance to BRAFi treatment. Increased expression of SerpinE2, JARID1B or PKH26 was associated with a slow-cycling phenotype [100]. Further, ALDH+ melanoma cells are more tumourigenic compared to ALDH−cells and differentiation capabilities [82]. In recent studies, NME1 demonstrated interesting characteristics depending on its expression level in melanoma cells [83,84]. High expression of NME1 promoted cell proliferation, growth of melanoma spheres and stem-cell-like features, whereas NME1low cells were more aggressive and metastatic while exhibiting neural crest-like features.

Peroxisome proliferator-activated receptor gamma coactivator 1-alpha (PGC1α), a transcriptional coactivator responsible for mitochondrial biogenesis and respiration [101], was heterogeneously expressed in melanoma [85,86]. PGC1α^high^ populations showed a proliferative phenotype, while PGC1α^low^ populations displayed a more metastatic phenotype [86]. More recently, a study showed that melanoma cells with higher metastatic potential have elevated levels of the MCT1 transporter, which takes in lactate. Both MCT1^high^ and MCT1^low^ cells were able to form tumours in NSG mice; however, MCT1^high^ cells formed more metastases compared to MCT1^low^ cells [87]. Emerging studies indicate that metabolic changes in melanoma cells are playing a part in melanoma metastasis. However, more studies need to be conducted to obtain enough information in determining how metabolic changes affect melanoma metastasis.

#### 3.2.3. Intratumoral Heterogeneity from Other Perspectives

##### Tumour Microenvironment

Bidirectional interactions between tumour cells and their microenvironment can also cultivate intratumoural heterogeneity. The tumour microenvironment consists of an extracellular matrix (ECM), stromal cells, fibroblasts and endothelial cells, amongst others, which play vital roles in tumour progression and metastasis. Tirosh and colleagues found that cancer associated fibroblasts (CAFs) expressed high levels of AXL in melanoma tumours that also expressed high AXL but apparent low levels of MITF, both markers of therapeutic resistance. In addition, genes expressed by CAFs were strongly correlated to infiltration of T cells into the tumour [54], which may impact patient response to immune checkpoint inhibitor therapy.

##### Immune Heterogeneity

Heterogeneous expression of melanoma differentiation antigens and immune infiltrates (CD4+ and CD8+ IHC staining) was observed in 11 melanoma samples [102]. Tissue microarray analysis was conducted on 21 samples (2–5 regions from each sample), where 11 samples showed a diverse presence of cytotoxic lymphocytes [103], indicating that there is no preferential localisation of immune cells within the tumour. Similarly, a diversified distribution of immune cells was observed in a tumour sample excised from a patient who was resistant to anti-PD-1 therapy [90]. Differential expression in immune profile (such as neoantigens, CD4, CD8 and Treg) was detected in a study of 33 synchronous metastases from 15 patients [35]. Of note, less than 8% of the total T-cell clones were shared between synchronous tumours, indicating significant immune heterogeneity observed intratumourally. It was found that highly heterogeneous tumours correlated with reduced anti-tumour immune cells, such as CD8+ T cells, follicular helper T cells and pro-inflammatory M1-like macrophages, and more immune suppressing cells, such as alternatively activated M2-macrophages [104]. Exposing B2905 murine melanoma cells to UVB irradiation, the main etiological cause of melanoma, considerably increased mutational loads from 2745 to 5479 exonic mutations, as well as increased the number of C > T transversions, an increase in the ultraviolet radiation mutation signature and immune heterogeneity [105]. The irradiated cells were more tumorigenic, but single-cell clones isolated from irradiated cells were less tumourigenic and more immunogenic. This could suggest that a reduction in the tumour mutational burden could reduce tumour aggressiveness and improve the treatment response.

## 4. Discoveries from Single-Cell Sequencing

Emerging technology, such as single-cell sequencing, enables melanoma heterogeneity to be scrutinised at both the intertumoural and intratumoural levels. Several studies have employed single-cell RNA sequencing (scRNA-seq) to identify subpopulations present within melanoma tumours based on gene expression signatures (Figure 1C). Despite the use of different approaches to identify cell states, there are similarities in the findings [54,106,107,108,109,110,111,112]. Thus far, up to seven distinct transcriptional melanoma cell states have been identified, although a consensus regarding the naming of these populations is yet to be reached (Table 2). Furthermore, a scRNA-seq was performed on tumours extracted from transgenic zebrafish with human *BRAF^V600E^* and *p53^−/−^* background [112]. There were three cell states identified; neural crest, mature melanocytic and stress-like, although the latter may have been a result of an artefact of the technology. Interestingly, the cell states showed a resemblance to those identified in human melanoma. Importantly, each of these states exhibits a unique transcriptional program based on the level of key transcription factors, and, therefore, phenotypic differences, including response to therapy. However, the states are likely to be highly dynamic and plastic, with a high level of switching between adjacent states based on microenvironmental signals. It has been suggested that several of these subpopulations may only arise following therapy. How these recently identified subpopulations fit with the previous work characterising heterogeneity within melanoma remains to be clarified and will be important for the field of melanoma biology moving forward.

Notably, Tirosh and colleagues also conducted scRNA-seq on tumour-infiltrating lymphocytes (TILs) collected from 15 melanomas, where they observed heterogeneous activation states of T cells within CD4+ and CD8+ populations [54]. In general, there were 28 differentially expressed gene signatures significantly correlated to highly exhausted populations across all tumours, which may shed light on immunotherapy resistance. The gene signatures, however, varied according to the cell lines, suggesting there are discrepancies due to the background of the tumours. Additionally, characterisation of CD8+ T cells from four patients who had received immunotherapy revealed 13 subpopulations based on RNA and surface protein expression [113] that were further categorised on the phenotypes as either ‘non-exhausted memory’ (T_NExM_) or exhausted (T_Ex_) cells. Notably, 83% of the T_Ex_ were tumour-specific compared to only 10% of T_NExM_. Consistent with that, tumour-specific T cell receptors (TCRs) were also enriched in mainly T_Ex_. Importantly, levels of circulating T_Ex_-related TCRs were associated with patients suffering worse disease outcomes. Although it is still debatable whether the T_Ex_ are dysfunctional, it is agreed that T_Ex_ have reduced capacity in reacting against cancer cells [114]. This study has depicted a plausible reason as to why immunotherapy is not effective in some patients and that generating non-exhausted T cells could improve treatment. Additionally, Kumar and colleagues investigated the interaction between CAFs, macrophages or tumour cells with their microenvironment by performing scRNA-seq on mouse xenografts [115]. Alterations in the chemokine family (including ligands and receptors) were found to be the most significant gene expression changes, followed by ECM components (collagen, fibronectin and integrin receptors). Nirschl and colleagues studied the enrichment of homeostasis differentiation signatures in tumour-associated mononuclear phagocytes by performing scRNA-seq on dendritic cells (DCs) and monocytes, both isolated from a lymph node melanoma metastasis [116]. The group identified two subpopulations, i.e., classical myeloid DCs and plasmacytoid DCs, and enrichment of interferon γ in the DCs.

ScRNA-seq has also been utilised in investigating the effect of treatment on melanoma cells. A study was conducted to elucidate the cell state transition of melanoma cells upon BRAFi treatment over a five-day course [117]. Initially, the cells expressed heterogeneous levels of various markers, such as MITF, MART-1, HIF1α and Ki-67. As the drug treatment started, metabolic regulators, as well as Ki-67, were downregulated, although the expression remained in a subpopulation, suggesting a slow-cycling phenotype was present in the Ki-67 negative cells. On day 3, elevated expressions of MART-1, resistance markers and metabolic regulators were observed. However, the MART-1 expression decreased by day 5, accompanied by an increase in EMT, enhanced expression of resistance markers and increased glucose uptake, indicating multiple simultaneous transitions were occurring, i.e., from cell differentiation to dedifferentiation (MART-1), transition of the cells toward mesenchymal phenotype, and, lastly, the activation of an acquired resistance mechanism. Interestingly, Shaffer and colleagues performed single-cell RNA fluorescence in in situ hybridisation (scRNA FISH) on 15 resistant markers in 8672 cells (one melanoma cell line) and detected a rare subpopulation of cells (0.2–2% of total cells) that expressed a high level of resistance markers prior to BRAFi treatment [92]. After BRAFi treatment for three weeks, the expression of resistance markers in the rare subpopulation was elevated, suggesting the existence of resistant cells in the population. The resistance mechanism does not appear to persist nor be inheritable as, when the rare subpopulation was cultured together with other cells over a week, the ratio of cells with one of the resistance markers, EGFR, reduced. The group then repeated scRNA FISH on 23 genes in 40,000 cells across four melanoma cell lines, four other cancer types and primary melanocytes to examine the phenomenon. Sporadic expression of the resistance markers was observed across all the cell lines, indicating that the phenomenon is not exclusive to cancer cells. Nevertheless, cell sorting according to resistant markers, such as AXL or NFGR, enriched the pre-resistant cells in some but not all cell lines, suggesting that the resistance mechanism is cell line-dependent and cancer-dependent. Similarly, a comparison of BRAFi resistance cell lines to the matched sensitive parental line found that the expression of *DCT*, *AXL* and *NRG1* increased with the development of resistance, indicating the potential existence of intrinsic resistant markers [115].

Advancements in technology have enabled a better understanding of melanoma biology, the effect of tumour microenvironment and the effect of treatment at a higher resolution and at a faster rate. Other than the examples that have been described here, there are many more interesting studies that provide invaluable insights in melanoma heterogeneity. It is expected that further studies investigating melanoma heterogeneity at the single-cell level will provide additional insights to the field.

## 5. Heterogeneity in Melanoma Progression

Tumour heterogeneity can also be observed during the transition from primary to metastatic melanoma. RASSF1A, a tumour suppressor that inhibits cell proliferation, was expressed at the superficial layer but not at the centre of the tumour due to promoter hypermethylation, indicating intratumoural heterogeneity as well as functionality of RASSF1A in different stages of melanoma [45]. Recently, a study has revealed that a circular RNA arising from a long non-coding RNA, *CDR1as*, was found abundantly in primary melanoma-derived cells but not matched metastatic-derived cells. *CDR1as* was epigenetically silenced, and that led to the activation of IGF2BP and upregulation of its targets, such as SNAI2 and MEF2C, which contributed to melanoma metastasis [118]. In a case study, WGS analysis revealed that, as a nevus of Ota progressed to primary melanoma and then to a recurrent tumour following initial excision, it acquired more mutations, including changes in the mutational status of genes, such as *BAP1* and *TP53,* and activation of RAS signalling [119]. IHC analysis showed that BAP1-expressing cells were predominantly found near the edges of the tumour, while p53-expressing cells were found in the proliferative region [119]. Interestingly, the p53-expressing clones were less aggressive, and the corresponding tumours were spontaneously rejected in mice. This could suggest that the existence of more clones/subpopulations (high intratumoural heterogeneity level), or intercellular communication between clones, is crucial in melanoma growth and progression.

## 6. Impact of Heterogeneity on Treatment Responses

The overall survival rate for metastatic melanoma patients has significantly improved owing to the development and advancement in targeted therapies and immunotherapies [120]. However, there is a large proportion of patients who do not benefit from these new treatments due to differential responses resulting from tumour heterogeneity. Resistance to BRAFi treatment is often associated with reactivation of the MAPK pathway through mechanisms such as *RAS* mutation, *BRAF* amplification and alternative splicing [121]. For example, a patient developed resistance to BRAFi treatment after subpopulations in the tumour switched from *NRAS^WT^/BRAF^V600E^* to *NRAS^G13R^/BRAF^V600E^*, thereby reactivating the MAPK pathway [44]. In a rare case of transplacental transmission of metastatic melanoma with *BRAF^V600E^* mutation, the mother was refractory to BRAFi treatment, whereas the baby was responsive to therapy [122]. Genomic analyses showed that there were at least two related but distinct subpopulations causing resistance, which later led to relapse. Notably, single-cell analysis of a panel of 16 melanoma cell lines and a primary cell line (epidermal melanocyte) found the status of markers, such as MITF, AXL and NGFR, impacted the BRAF/MEKi treatment [123]. AXL^high^ cells were more resistant to various treatments in comparison to MITF^high^ or NGFR^high^ cells. Resistant AXL^high^ cells exhibited partial MAPK pathway inhibition with high Ki67 expression, while both MITF^high^ and NGFR^high^ cells demonstrated considerable inhibition of the MAPK pathway, suggesting that the MAPK pathway is crucial for AXL^high^ cells but not MITF^high^ and NGFR^high^ cells. Co-treatment of BRAF/MEKi with different epigenetic modulators in the cells revealed selective sensitivities presented by different populations of cells [123]. Several slow-cycling subpopulations, such as MITF^low^/NF-κB^high^/AXL^high^, are also correlated to reactivation of the MAPK pathway [68,69]. Importantly, elevated expression of AXL was observed following BRAFi and MEKi (either single or combination) treatments in patient-derived xenografts (PDXs) [99]. To better understand the mechanisms behind resistance, Shaffer and colleagues compared the expression of resistance markers in pre-resistant cells and BRAFi-treated cells [92]. Interestingly, the number of resistance markers expressed increased from 72 in the pre-resistant cells to 600 (of the 1456 genes tested) in the post-drug-treatment cells by transcriptome analysis. Only 33 differentially accessible sites were discovered by comparing the non-resistant and pre-resistant cells versus more than 10,000 differentially accessible sites between pre- and post-drug-treatment using ATAC-seq, suggesting minute variations between the non-resistant and pre-resistant cells, and that the treatment induced phenotype switching resistance. Yet, whether the 33 differentially accessible sites between non-resistant and pre-resistant cells are sufficient to provoke a remarkable effect in drug resistance is arguable. In another study, resistance-conferring genetic alterations were evidenced in PDXs established from treatment-naïve patient samples, suggesting the existence of intrinsic resistant populations in melanoma patients [124]. Nevertheless, acquired resistance was observed in some PDXs too, indicating variabilities in causations of resistance to treatments. Meanwhile, other slow-cycling populations, such as JARID1B^high^ cells, CD133+, ABCB5+ and NGFR+ cells, have also been implicated in resistance to BRAFi and other anti-cancer drugs [49,76,78,125,126]. Furthermore, the PIK3CA^E545K^ mutation was found to confer resistance to combined MEKi + CDK4i in a melanoma patient [127]. Nonetheless, aberrant expression of methyltransferases in melanomas could affect the sensitivity of melanoma cells to chemotherapy [128].

In addition, not only did pre-existing NGFR^high^ melanoma cells show reduced sensitivity to the combination of BRAFi and MEKi but the cells were also resistant to antigen-specific cytotoxic CD8+ T cells [129]. Consistent with this finding, patients whose tumours are NGFR^high^ were not responsive to anti-PD-1 and/or a combination of anti-PD-1 and anti-CTLA-4 therapy [129]. Furthermore, acquired *JAK1* and *JAK2* loss-of-function mutations were found in tumours after patients developed resistance to anti-PD-1 therapy [130]. The mutations were discovered by comparing the whole-exome sequencing data of tumour samples before treatment and after relapse. The cells with *JAK1* mutations were resistant to interferon α, β and γ-induced growth arrest, while the cells with *JAK2* mutations were insensitive to interferon γ-induced growth arrest. The same study also revealed the mutation of beta-2-microglobulin (B2M) in another patient’s baseline and matched relapse samples, where the loss of B2M reduced the immune-cell recognition of cancer cells. Nonetheless, there are ambiguous responses to immunotherapy due to tumour heterogeneity. For instance, data from a mouse study suggested that tumours with a low level of intratumoural heterogeneity may be more immunogenic and, hence, have an improved overall survival rate [105]. Lin and colleagues found that tumours with high heterogeneity were associated with decreased immune cell infiltration, immune response activation and overall patient survival [104]. In contrast, a positive correlation was observed between high mutational load and enhanced T cell infiltration in patient samples [35], indicating that tumours with a high mutational load may be more immunogenic. Despite the inconsistencies observed, it is possible that the identity of the genes mutated is potentially more pertinent in determining the treatment response regardless of the amount of mutational load [131].

Altogether, studies have shown that drug treatment could trigger secondary mutations in tumour cells, which lead to resistance. It is also evident that some tumour cells harbour pre-existing resistance, conferring genetic properties that reduce sensitivities to treatments and later lead to treatment resistance. The heterogeneous tumour environment similarly contributes to immunotherapy treatment resistance. The development of immunotherapy has greatly transformed the treatment for advanced melanoma and dramatically increased the overall survival rate for melanoma patients. However, there are still nearly 40% of advanced melanoma patients that do not respond initially to immunotherapies, with adverse side effects observed in 60% of the patients [132]. The reason why some patients respond well while others do not respond at all is still unclear, although it is clear that heterogeneity at the tumour and immune levels plays an important role.

## 7. Conclusions and Implications

Heterogeneity within melanoma was initially described by Norris, an attentive general practitioner, in 1820 [133]. Since that time, we have gained significantly more discerning information regarding melanoma development and tumour heterogeneity, particularly at the molecular level. However, many aspects of the development and role of heterogeneity remain unclear. For example, the true identity of each of the subpopulations within the patient tumour remains to be clarified. Further, the specific roles of communication between different subpopulations within the tumour and the various biological components within the tumour microenvironment (ECM and immune cells) are not well characterised. As more subpopulations develop in the tumour, the more challenges patients must face, such as tumour growth, metastasis, efficacy of the therapeutic treatments and, finally, resistance to treatment. These open questions must serve as a reminder that much more needs to be done in order to fill in the missing pieces and truly understand the role of heterogeneity in melanoma metastasis and treatment resistance in order to effectively tackle the disease. Effectively targeting the tumour heterogeneity in melanoma could lead to better therapeutic options and better outcomes for patients.

## Figures and Tables

**Figure 1 cancers-14-03030-f001:**
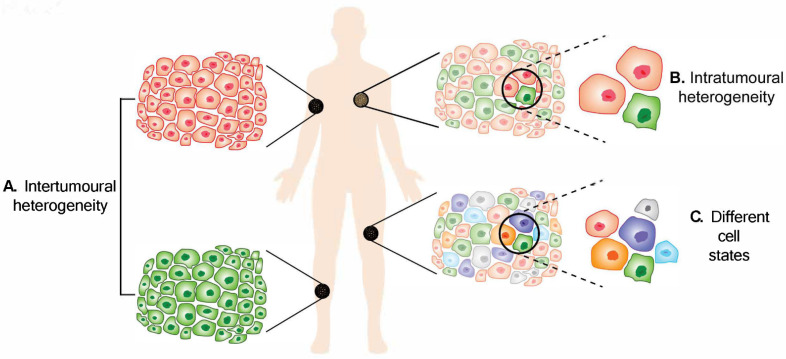
Melanoma heterogeneity. (**A**) Intertumoural heterogeneity refers to differences between lesions in a patient, while (**B**) intratumoural heterogeneity describes the differences between cells or subpopulations of cells within a tumour. Advancement in technology has revealed (**C**) different cell states in melanoma tumours.

**Table 2 cancers-14-03030-t002:** Different cell states proposed in different findings.

Ref.	Cell States							Samples
	Cycling/Proliferative		Intermediate		Non-Cycling			
[54]	MITF^high^				AXL^high^			Patient tumour samples (advanced stage)
[106]	Proliferation		Pigmentation		Stromal			Patient-derived cultures (advanced stage)
[107]	C4 melanocytic		C3 transitory		C2 neural crest-like		C1 undifferentiated	Human melanoma cell lines (53)
[108]	Hyper-differentiated	Melanocytic	Intermediate		Starved	Neural crest stem-cell-like	Undifferentiated	PDXs from advanced-stage patients
[109]	State #1 (high cyclin D1, ERBB3, STAT3/5, ATF1, ATF4, MITF & β-catenin; low c-JUN, Axl & EGFR)		State #2 (high ERBB3 Axl & c-JUN; low MET, RELB, E2F1, BIM, ULK1, SMAD1/9 & XIAP)		State #3 (high Axl, c-JUN, E2F1, WEE1, c-MET & EGFR; low MITF, ERBB3 and SMAD9)		State #4 (low MITF/RTK expression & suppressed cell-death-related gene expression)	Human melanoma cell lines (1205Lu, 1205LuR, WM164 & WM164R)
[110]	Melanocytic		Intermediate		Mesenchymal-like			Patient-derived cultures (9) and human melanoma cell line (A375)
[111]	C2 high cycling (G1/S)	C4 high cycling (G2/M)	C5 translation	C7 reactivation of MAPK	C6 pluripotent	C1 neural crest-like	C3 slow cycling, stroma-like	Human melanoma cell lines (A375)
[112]	Mature melanocytic		Stress-like		Neural crest			Human melanoma-like tumour from transgenic zebrafish

For more details, please refer to original articles.

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
