# Peer review of "Heterogeneity in Melanoma"

_cancers, 2022, doi:10.3390/cancers14123030_

Round 1

Reviewer 1 Report

The manuscript “Heterogeneity in Melanoma” is a review regarding the melanoma tumour heterogeneity, and what is known about its impact on disease progression and treatment response. The manuscript is well written and easy to read. I appreciate the work performed by the authors and I think that the manuscript is of interest for the readers since points out the actual implications of melanomas heterogeneity.

However, there are some concerns that the author should address before the manuscript could be considered for publication:

1.       Line 32: “Cutaneous melanoma arises when melanocytes, the cells that produce melanin and provide ultraviolet (UV) protection to the skin divide uncontrollably”. This sentence is misleading since the ability to divide uncontrollably is only one of the features that characterize a malignant transformation. Please correct it.

2.       In the introduction section authors report that exposure to UV radiation results in mutations in DNA implicated in melanoma development. However, they should specify that UVB not only induces the formation of cyclobutane pyrimidine dimers and pyrimidine-pyrimidone(6–4) products but can also trigger the formation of SSBs and DSBs on DNA which promote the carcinogenesis. Furthermore, UVA irradiation can induce the formation of ROS, and this ROS overproduction can damage DNA (PMID: 35453297).

3.       Line 66: authors interestingly report that differential methylation was detected between three paired primary and metastatic cell lines. A possible connection between these findings and the aberrant expression of methyltransferases in melanomas should be discussed, since it has been proposed as a mechanism of malignant progression (PMID: 23455543; PMID: 34638427).

4.       Paragraph 3: authors should create a table to summarize data presented in this section.

5.       The table 1 is confused. It requires to be modified and present data more clearly.

6.       Line 445: “there are a large proportion of patients” should be modified in “there is a large proportion of patients”.

7.       In paragraph 6 authors should discuss the possible impact deriving from aberrant expression of methyltransferases in melanomas on chemoresistance, since they impact the genomics (PMID: 34018676).

8.       In the section “Summary and Implications” authors should highlight the possible application of the presented findings, and the perspectives of future researches to be carried on, in order to improve the management of melanoma.

Author Response

We thank the Reviewers for their positive and considered comments, which we feel have lead to great improvements in the manuscript.  We have addressed all of the Reviewer’s comments in the updated resubmission.  All changes to the original manuscript have been highlighted in the updated version for ease of review.  The additional references requested by the Reviewers have been included at the end of the reference list.  This will alter the numbering of the references; please don’t hesitate to let me know if this is a problem.  Please see the attached point-by-point response to the Reviewers comments below.

Reviewer 1:

The manuscript “Heterogeneity in Melanoma” is a review regarding the melanoma tumour heterogeneity, and what is known about its impact on disease progression and treatment response. The manuscript is well written and easy to read. I appreciate the work performed by the authors and I think that the manuscript is of interest for the readers since points out the actual implications of melanomas heterogeneity.

However, there are some concerns that the author should address before the manuscript could be considered for publication:

  1. Line 32: “Cutaneous melanoma arises when melanocytes, the cells that produce melanin and provide ultraviolet (UV) protection to the skin divide uncontrollably”. This sentence is misleading since the ability to divide uncontrollably is only one of the features that characterize a malignant transformation. Please correct it.

We thank the Reviewer for the comment. We have made correction to the sentence, changed it from “divide uncontrollably” to “undergo malignant transformation” (page 1, line 32-33).

  1. In the introduction section authors report that exposure to UV radiation results in mutations in DNA implicated in melanoma development. However, they should specify that UVB not only induces the formation of cyclobutane pyrimidine dimers and pyrimidine-pyrimidone(6–4) products but can also trigger the formation of SSBs and DSBs on DNA which promote the carcinogenesis. Furthermore, UVA irradiation can induce the formation of ROS, and this ROS overproduction can damage DNA (PMID: 35453297).

We appreciate suggestion from the Reviewer.  We now include this additional information. Please see page 1, lines 37-40 for the updated text.

  1. Line 66: authors interestingly report that differential methylation was detected between three paired primary and metastatic cell lines. A possible connection between these findings and the aberrant expression of methyltransferases in melanomas should be discussed, since it has been proposed as a mechanism of malignant progression (PMID: 23455543; PMID: 34638427).

We appreciate the suggestion from the Reviewer. We now include reference to the aberrant expression of methyltransferases in melanoma.  Please see lines 78-79.

  1. Paragraph 3: authors should create a table to summarize data presented in this section.

With respect to the Reviewer, the information contained in Section 3 is too great to be summarised in a table.  The section refers to over 80 different studies, with heterogeneity of differing origins discussed in detail.  We would be afraid that a summary table would not be sufficient in conveying the data.

  1. The table 1 is confused. It requires to be modified and present data more clearly.

We have now modified the table and added additional information so that the groups are easier to understand (Table 1).  We include an updated table in the revision.

  1. Line 445: “there are a large proportion of patients” should be modified in “there is a large proportion of patients”.

We thank the Reviewer for the comment. The sentence has been corrected (line 454).

  1. In paragraph 6 authors should discuss the possible impact deriving from aberrant expression of methyltransferases in melanomas on chemoresistance, since they impact the genomics (PMID: 34018676).

The additional information requested by the Reviewer has been added.  Please see lines 494-496 for the additional text.

  1. In the section “Summary and Implications” authors should highlight the possible application of the presented findings, and the perspectives of future researches to be carried on, in order to improve the management of melanoma.

Additional information has been added, please see line 542-543.

Reviewer 2:

The manuscript titled “Heterogeneity in Melanoma” reviewed the importance of tumor heterogeneity of melanoma contributing to tumor regression and metastasis. I understand one review article could not cover all the corners of the field. However, at least some important or similar works should be mentioned to make a good review. The followings are some concerns and comments have been pointed out that the authors may want to consider.

  1. I think the authors might forget to mention at least the similar review articles: a) /Grzywa TM, Paskal W, WÅ‚odarski PK. Intratumor and Intertumor Heterogeneity in Melanoma. Transl Oncol. 2017 Dec;10(6):956-975. doi: 10.1016/j.tranon.2017.09.007. Epub 2017 Oct 24. PMID: 29078205; PMCID: PMC5671412./; b)/Ito T, Tanaka Y, Murata M, Kaku-Ito Y, Furue K, Furue M. BRAF Heterogeneity in Melanoma. Curr Treat Options Oncol. 2021 Feb 8;22(3):20. doi: 10.1007/s11864-021-00818-3. PMID: 33558987./;

We thank the Reviewer for the suggestion.  We have now amended this section.  Please see page 2, line 54-56 for the updated text.

  1. Line 53 Figure 1: In the image, the word should be “tumoural”.

We appreciate the Reviewer’s careful reading.  We have amended Figure 1 in the revision.

  1. Line 360 Table 1: I’d suggest the authors provide some more details from the references to the samples category. For example, what are the human melanoma cell lines? Primary or metastasis from patients’ tissue?

We thank the Reviewer for the comment. We have now made amendments within the table to clarify the sample origins within the listed studies.  Please see Table 1 for the update.

  1. References: Please release authors’ names (as much as you can based on the journal guideline) for easier checking instead of only one author listed for each of them.

The reference list has been formatted by the Editorial office to reflect in-house requirements.  We are unable to alter the referencing.

Reviewer 3:

The incidence rate of melanoma increased and melanoma is a cancer that exhibits one of the most heterogeneous features. These levels of tumor heterogeneity hinder accurate diagnosis and effective treatment. This study described the melanoma variety and the tumor heterogeneity.

In introduction it could be useful to introduce the Heterogeneity in tumors.

We thank the Reviewer for the comment.  We direct the reader to lines 50-55, and to Figure 1 where the different aspects of heterogeneity in tumours is addressed.

Please reported if Heterogeneity was found in hematopoietic malignancies or in solid human tumors.

We appreciate the Reviewer’s question.  We have now addressed the presence of heterogeneity in hematopoietic and solid malignancies (all cancers) and provided a reference to an excellent review.  Please see line 49.

Please insert classification of cutaneous melanoma about genomic characteristics.

We now address the Reviewer’s request by including reference to the 4 generally recognised cutaneous melanoma groups.  Additional information has been added in the text; please see lines 41-42.

Thank you for the classification of level of heterogeneity (intratumor/ intertumoral/interpatient). The Heterogeneity bases are genome, trascriptome/Proteome and epigenome. At the end of paragraph 3, Please insert summary table about the levels of heterogeneity described and the three basis of heterogeneity (genetic, genes' expression and epigenetic) reporting reference and results

We thank the Reviewer for the suggestion.  Indeed, each of the sub-headings of Section 3 highlight the suggestion of the Reviewer: genetic versus non-genetic basis of heterogeneity, with the non-genetic contribution further broken into epigenetic, phenotypic, immune and extracellular sources of heterogeneity.  Please see Section 3, and the subheading contained therein.

Reviewer 2 Report

The manuscript titled “Heterogeneity in Melanoma” reviewed the importance of tumor heterogeneity of melanoma contributing to tumor regression and metastasis. I understand one review article could not cover all the corners of the field. However, at least some important or similar works should be mentioned to make a good review. The followings are some concerns and comments have been pointed out that the authors may want to consider.

1.   I think the authors might forget to mention at least the similar review articles: a) /Grzywa TM, Paskal W, WÅ‚odarski PK. Intratumor and Intertumor Heterogeneity in Melanoma. Transl Oncol. 2017 Dec;10(6):956-975. doi: 10.1016/j.tranon.2017.09.007. Epub 2017 Oct 24. PMID: 29078205; PMCID: PMC5671412./; b)/Ito T, Tanaka Y, Murata M, Kaku-Ito Y, Furue K, Furue M. BRAF Heterogeneity in Melanoma. Curr Treat Options Oncol. 2021 Feb 8;22(3):20. doi: 10.1007/s11864-021-00818-3. PMID: 33558987./;

2.      Line 53 Figure 1: In the image, the word should be “tumoural”.

3.      Line 360 Table 1: I’d suggest the authors provide some more details from the references to the samples category. For example, what are the human melanoma cell lines? Primary or metastasis from patients’ tissue?

4.      References: Please release authors’ names (as much as you can based on the journal guideline) for easier checking instead of only one author listed for each of them.

Author Response

(The authors gave the same response as above.)

Reviewer 3 Report

The incidence rate of melanoma increased and melanoma is a cancer that exhibits one of the most heterogeneous features. These levels of tumor heterogeneity hinder accurate diagnosis and effective treatment. This study described the melanoma variety and the tumor heterogeneity.

In introduction it could be useful to introduce the Heterogeneity in tumors.

Please reported if Heterogeneity was found in hematopoietic malignancies or in solid human tumors.

Please insert classification of cutaneous melanoma about genomic characteristics. 

Thank you for the classification of level of heterogeneity (intratumor/ intertumoral/interpatient). The Heterogeneity bases are genome, trascriptome/Proteome and epigenome. At the end of paragraph 3, Please insert summary table about the levels of heterogeneity described and the three basis of heterogeneity (genetic, genes' expression and epigenetic) reporting reference and results

Author Response

(The authors gave the same response as above.)

Round 2

Reviewer 2 Report

I do not have any further concerns now, just please double-check to homogenous the format throughout the manuscript again before publication.  Good luck.

Author Response

Reviewer 1

No comment

No response required.

Reviewer 2

I do not have any further concerns now, just please double-check to homogenous the format throughout the manuscript again before publication.  Good luck.

We thank the Reviewer for the previous feedback.  We have carefully read through and edited the manuscript to ensure consistency in the formatting.

Reviewer 3

We thank the authors for the underlying the references and the lines about the questions. The manuscript is not very different from first version.

I understand one review article could not cover all the corners of the field. However, at least some important or similar works should be mentioned to make a good review.

We draw the attention of the Reviewer to the previous response to Reviewer 2, where we included reference to two similar works, specifically:

  1. Grzywa, T.M., W. Paskal, and P.K. Wlodarski, Intratumor and Intertumor Heterogeneity in Melanoma. Transl Oncol, 2017. 10(6): p. 956-975.
  2. Ito, T., et al., BRAF Heterogeneity in Melanoma. Curr Treat Options Oncol, 2021. 22(3): p. 20.

Please see line 56 for the reference to the previous reviews.

As suggested in first round, the authors may want to consider and implemente the aspects about the Heterogeneity in tumors, classification of cutaneous melanoma about genomic characteristics, not only underlying the references and the lines about the questions.

We would like to draw the attention of the Reviewer to the previously added sentences and references as requested in the last review.  In this revision, we now outline that heterogeneity is found in both hematological and solid tumours; please see line 49 for the changes.  We feel that reference to an excellent review covering this topic is appropriate, as the focus of the current manuscript is specifically on heterogeneity in cutaneous melanoma.  Please see lines 40 to 42 for the classification of cutaneous melanoma subtypes, and the genetic variations that define them.

We thank the authors for the subheading of section 3 but a summary table about the levels of heterogeneity described and the three basis of heterogeneity (genetic, genes' expression and epigenetic) reporting reference and results is useful.

Please insert summary table at the end of paragraph 3.

We now include a new Table to comply with the Reviewer’s request.  We have summarized all the populations discussed in Section 3, and broken the groupings in “Genetic”, “Epigenetic” and “Phenotypic” as suggested by the Reviewer.

Reviewer 3 Report

We thank the authors for the underlying the references and the lines about the questions. The manuscript is not very different from first version.

I understand one review article could not cover all the corners of the field. However, at least some important or similar works should be mentioned to make a good review.

As suggested in first round, the authors may want to consider and implemente the aspects about the Heterogeneity in tumors, classification of cutaneous melanoma about genomic characteristics, not only underlying the references and the lines about the questions.

We thank the authors for the subheading of section 3 but a summary table about the levels of heterogeneity described and the three basis of heterogeneity (genetic, genes' expression and epigenetic) reporting reference and results is useful.

Please insert summary table at the end of paragraph 3.

Author Response

(The authors gave the same response as above.)

Round 3

Reviewer 3 Report

I have no questions and suggestion for the modified study.